# Utilizing co-abundances of antimicrobial resistance genes to identify potential co-selection in the resistome

Hannah-Marie Martiny,[1] Patrick Munk,[1] Christian Brinch,[1] Frank M. Aarestrup,[1] M. Luz Calle,[2] Thomas N. Petersen[1]

**ABSTRACT** The rapid spread of antimicrobial resistance (AMR) is a threat to global health, and the nature of co-occurring antimicrobial resistance genes (ARGs) may cause collateral AMR effects once antimicrobial agents are used. Therefore, it is essential to identify which pairs of ARGs co-occur. Given the wealth of next-generation sequencing data available in public repositories, we have investigated the correlation between ARG abundances in a collection of 214,095 metagenomic data sets. Using more than $6.76 \cdot 10^8$ read fragments aligned to acquired ARGs to infer pairwise correlation coefficients, we found that more ARGs correlated with each other in human and animal sampling origins than in soil and water environments. Furthermore, we argued that the correlations could serve as risk profiles of resistance co-occurring to critically important antimicrobials (CIAs). Using these profiles, we found evidence of several ARGs conferring resistance for CIAs being co-abundant, such as tetracycline ARGs correlating with most other forms of resistance. In conclusion, this study highlights the important ARG players indirectly involved in shaping the resistomes of various environments that can serve as monitoring targets in AMR surveillance programs.

**IMPORTANCE** Understanding the collateral effects happening in a resistome can reveal previously unknown links between antimicrobial resistance genes (ARGs). Through the analysis of pairwise ARG abundances in 214K metagenomic samples, we observed that the co-abundance is highly dependent on the environmental context and argue that these correlations can be used to show the risk of co-selection occurring in different settings.

**KEYWORDS** metagenomics, correlation, network analysis, compositional data analysis, co-abundances, antimicrobial resistance, microbiome

Antimicrobial resistance (AMR) has become a global health threat, with ramifications not only for modern medicine but also for agriculture and environmental health (1–3). It is widely acknowledged that the misuse of antimicrobials has accelerated the dissemination and prevalence of antimicrobial resistance genes (ARGs) (4). Most attempts to reduce the burden of AMR have focused on reducing the use of single classes of antimicrobial agents considered of critical importance (5). Despite considerable efforts in various settings, such as livestock, these regulations have not significantly reduced the spread of ARGs (6, 7). It has been shown that after banning the use of specific antimicrobials, resistance to that antibiotic will still be prevalent in the environment (8). Indirect co-occurrence of ARGs has also been shown to happen under selective antibiotic pressure (7, 9–11). We have also recently observed that changes in the selective pressure of even a single antimicrobial agent influence several ARGs in pig metagenomes (11). However, there is also evidence that adding mobile ARGs to a microbiome increased its stability, whereas if the ARGs were encoded in the chromosome, the stability decreased (12). The origin of the interplay between the

Address correspondence to Hannah-Marie Martiny, hanmar@food.dtu.dk.

The authors declare no conflict of interest.

See the funding table on p. 10.

microbiome and resistome, however, remains largely unresolved, as there are many ecological, functional, and evolutionary properties unaccounted for. It is unclear whether the co-selection of ARGs happens due to genetic linkage or inter-species interactions.

Studies that focused on studying the link between antimicrobial usages (AMUs) and the prevalence of AMR have typically only focused on the effects of using one antimicrobial agent and the impact on developing resistance to that agent, ignoring the overall effect on the abundance of ARGs in the environment. This effect on the resistome must be understood better to prevent these collateral damage effects from happening (13, 14). Co-occurrence of microbes has been evaluated in soil (15, 16) and marine environments (17, 18), whereas ARG co-occurrences have been studied in sewage sludge (19), freshwater (20), marine (21), swine (8), and cattle (22). However, most of these studies have only been evaluated on a smaller scale and do not always use the same methods. There is currently a large collection of next-generation sequencing data sets from metagenomic samples available in public repositories, providing an excellent resource to quantify the prevalence of ARGs by analyzing the read abundances (23–26).

In this study, we have analyzed the co-abundance of sequencing reads aligned to acquired ARGs to assess how resistance to one specific antimicrobial agent is linked to the abundance of another class of antimicrobial. With a collection of 214,095 metagenomic data sets from public repositories (27), we examined the correlation between pairwise ARG read abundances with a compositional approach (28, 29) using SparCC (*Spar*se *C*orrelations for *C*ompositional data) (30). Our results demonstrated that many ARG pairs interact but are highly specific to the environment. We believe that these interactions provide another way to study how ARGs are being co-selected independently of the microbial context, and the findings can be used to design targeted interventions to limit the spread of AMR.

## MATERIALS AND METHODS

### Data collection and pre-processing

We have previously described in detail the process of downloading and analyzing 214,095 metagenomic samples (25, 27), but in brief, we downloaded raw sequencing reads corresponding to 442 Tbp from metagenomic samples deposited in the European Nucleotide Archive (31) that were uploaded between 1 January 2010 and 1 January 2020 and had at least 100,000 reads and were shotgun sequenced. The raw sequencing reads were quality-checked with FASTQC v.0.11.15 (https://www.bioinformatics.babraham.ac.uk/projects/fastqc/) and trimmed with BBduk2 36.49 (32). The trimmed reads were then globally aligned using the k-mer alignment tool (KMA) (33) 1.2.21 against two reference sequence databases: ResFinder (34) (downloaded 25 January 2020) and Silva (35) (version 38, downloaded 16 January 2020). ResFinder is a database of 3,085 acquired ARGs, whereas Silva is a 16S/18S rRNA database of 2,225,272 sequences.

### Homology-based clustering of ResFinder sequences

With the possibility of detecting up to $C(3,085, 2) = \frac{3,085!}{(2!(3,085 - 2)!)} = 4,757,070$ pairwise interactions between the 3,085 ARGs of ResFinder, we decided to cluster similar reference sequences together to reduce the number of possible combinations. Using USEARCH (v11.0.667), we did a homology-based clustering that grouped ARG sequences into 90% identity groups, producing a total of 716 clusters (36, 37). For each gene member of an ARG cluster, the read counts for the gene were first adjusted by the gene sequence length, and then all length-adjusted counts were summed together (38). These new ARG cluster counts are used in the correlation analysis described in the next section and are referred to as simply ARGs in the remainder of the manuscript.

## Calculating relative abundances

For a category label, we calculated the relative abundance of fragment counts assigned to different genes or classes as:

$$\text{Relative abundance}(x) = \frac{\kappa}{\sum n_i} n_i,$$

where $x$ is the label, $n_i$ is the count of read fragments assigned to gene $i$, and $\kappa = 100$ is a scaling constant (38).

## Inferring pairwise correlations with SparCC

The SparCC algorithm was used to obtain correlations using pairs of log-ratio transformed ARG read counts to infer linear Pearson correlations (30). SparCC obtains linear Pearson correlations and $P$ values through an iterative approach that adjusts for spurious correlations and lowers the false discovery rate. The ARG–ARG correlations were inferred as the average over 50 iterations, and one-sided pseudo-$P$ values were obtained through a bootstrapping procedure of 100 rounds. In each bootstrapping round, the input count matrix was shuffled, and correlations were averaged over 10 iterations to infer one-sided $P$ values to test whether the correlations for the observed data were statistically significant. Correlation $\geq 0.6$ with $P \leq 0.01$ were selected for further analysis. We implemented SparCC to run on GPUs on the Danish National Supercomputer for Life Sciences (https://www.computerome.dk).

We ran SparCC on the entire data set of the 214K metagenomic samples and subsets of samples grouped by sampling host and environment, where at least 800 samples existed. Due to inconsistent labeling of the sampling sources, we made new source groups, as shown in Table 1. We only consider genes for SparCC analysis that are present in at least 10 samples with a minimum read fragment count of 50. In total, 11 different correlation matrices were made, that is, one for each of the source groups listed in Table 1. The correlation networks were visualized in R 4.1.0 (39) with packages igraph (40), qgraph (41), and ggraph (42) using the Fruchterman–Reingold layout algorithm (43).

## Network comparisons

The topology of the different networks is described using different metrics: the number of nodes ($N$) and edges ($E$), the global clustering coefficient, network density, edge density, and the number of components. The global clustering coefficient, or the graph

**TABLE 1** Grouped labels for hosts and environments[a]

| Source group | Sampling source label(s) | Number of samples | Number of samples with ARGs |
|---|---|---|---|
| Air | Air metagenome | 914 | 870 |
| Chicken | *Gallus gallus* (1,219), chicken gut metagenome (4) | 1,223 | 1,215 |
| Cow | *Bos taurus* (872), cow dung metagenome (14) | 886 | 824 |
| Dog | *Canis lupus familiaris* | 3,439 | 3,182 |
| Freshwater | Freshwater metagenome | 4,494 | 585 |
| Human | *Homo sapiens* | 95,003 | 57,239 |
| Marine | Marine metagenome | 30,002 | 5,444 |
| Mouse | *Mus musculus* (1,462), mouse metagenome (50), mouse gut metagenome (2,435) | 3,947 | 3,909 |
| Pig | *Sus scrofa* (673), *Sus scrofa domesticus* (355), pig metagenome (2,129), pig gut metagenome (72) | 3,229 | 3,461 |
| Soil | Soil metagenome | 6,533 | 2,822 |
| All | | 214,095 | 119,206 |

[a]The parenthesis after each sampling source label denotes the number of samples assigned to that label if the group consisted of multiple labels.

transitivity, measures the density of node triplets in the network (44). The network density is calculated as $2E/N(N-1)$ as given in Parente et al. (2018) (45). Edge density is the number of edges over the number of possible edges (46). The average correlation between ARGs of two antimicrobial classes was calculated using Fisher's *z*-transformation on the correlation values, averaging the *z*-scores and converting it back to a correlation score with the inverse Fisher transformation (47).

## RESULTS

This study investigated the correlation of pairwise ARG abundances across a set of 214,095 metagenomic sequencing data sets. This collection represents a highly diverse set of publicly available metagenomic samples collected between 2000 and 2020 from locations all over the world, where the sequencing data sets consist of short-read reads produced on Illumina platforms (Fig. S1). We observed that not all samples contained sequence read fragments aligned to ARGs, so we only included those that did in the correlation analyses (*n* = 119,206; Table 1). Looking at the various sampling sources, read fragments that aligned to ARGs' conferring resistance to tetracycline and beta-lactams were very common in host-associated sources, whereas phenicol resistance was more frequently observed in environmental sources (freshwater, marine, and soil) (Fig. 1). *catA1* was the most dominant gene in the environmental samples, especially in marine samples, whereas various *tet* genes had high relative abundances in livestock or human samples: *tet(W)* in chicken and pig samples, and *tet(Q)* in cow, human, and pig metagenomes. *blaTEM-52B* accounted for more than 30% of the read fragments assigned to ARGs in air metagenomes (Fig. S2).

### Balancing the sparsity and network complexity

Initially, we inferred correlations with SparCC on all samples that had at least one read fragment aligned to an ARG. We observed that most ARGs were found to correlate with each other in all environments, even in cases where there was only one read fragment per reference (Table S1). Across all samples, we observed template hits ranging from 1 to 252 with a median of 20 and fragment counts spanning 1 to $4.5 \cdot 10^7$, with a median of 880 fragment counts. To balance the sparsity and network complexity, we introduced a filtering step. This step determined whether an ARG should be included or excluded based on a minimum number of fragment counts and a minimum number of samples supporting hits to that ARG template. Among the 716 ARGs considered, we noticed that without any filters, SparCC indicated correlations even with very low read fragment counts. Therefore, we required that for an ARG to be included, it had to have a minimum fragment count of 50 across at least 10 samples (Table S1).

### Analysis of large-scale metagenomic correlation networks

Based on our filter settings, we constructed a global network using the correlation coefficients for the entire collection of metagenomes, with each node representing an ARG and each edge representing a pairwise ARG connection (correlation ≥ 0.6, *P* < 0.01; Fig. S3). The global network, nicknamed "all," contained 225 ARGs connected through 2,344 correlation edges (Fig. 2A). As this all network was hard to interpret due to the many highly interconnected ARGs, we also inferred pairwise ARG correlations in specific sampling groups (Table 1). The genes that were part of these networks were found to correlate with varying degrees of strength (Fig. 2A and B). For example, the human network contained many correlation coefficients, but most were less than 0.8 (Fig. 2C). Another example is the marine network, where only a few ARGs were found to correlate but with values above 0.9 (Fig. 2B and C). Despite the networks reflecting the composition of the various environments, we still observed overlaps between which ARGs were found to correlate. One example is that all the correlations inferred from the pig metagenomes also existed in the human metagenomic network (Fig. 2D).

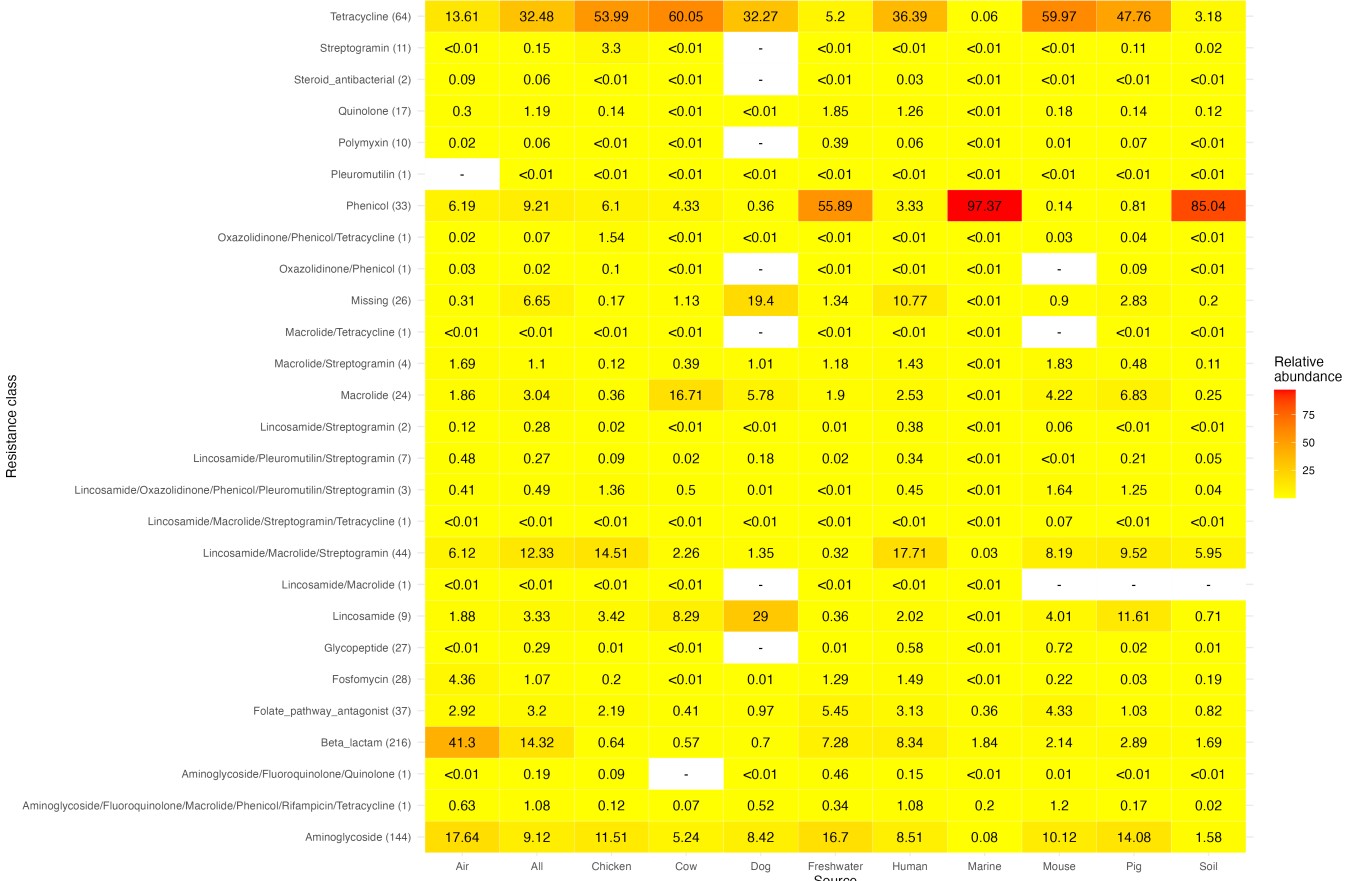

| Resistance class | Air | All | Chicken | Cow | Dog | Freshwater | Human | Marine | Mouse | Pig | Soil |
|---|---|---|---|---|---|---|---|---|---|---|---|
| Tetracycline (64) | 13.61 | 32.48 | 53.99 | 60.05 | 32.27 | 5.2 | 36.39 | 0.06 | 59.97 | 47.76 | 3.18 |
| Streptogramin (11) | <0.01 | 0.15 | 3.3 | <0.01 | - | <0.01 | <0.01 | <0.01 | <0.01 | 0.11 | 0.02 |
| Steroid_antibacterial (2) | 0.09 | 0.06 | <0.01 | <0.01 | - | <0.01 | 0.03 | <0.01 | <0.01 | <0.01 | <0.01 |
| Quinolone (17) | 0.3 | 1.19 | 0.14 | <0.01 | <0.01 | 1.85 | 1.26 | <0.01 | 0.18 | 0.14 | 0.12 |
| Polymyxin (10) | 0.02 | 0.06 | <0.01 | <0.01 | - | 0.39 | 0.06 | <0.01 | 0.01 | 0.07 | <0.01 |
| Pleuromutilin (1) | - | <0.01 | <0.01 | <0.01 | <0.01 | <0.01 | <0.01 | <0.01 | <0.01 | <0.01 | <0.01 |
| Phenicol (33) | 6.19 | 9.21 | 6.1 | 4.33 | 0.36 | 55.89 | 3.33 | 97.37 | 0.14 | 0.81 | 85.04 |
| Oxazolidinone/Phenicol/Tetracycline (1) | 0.02 | 0.07 | 1.54 | <0.01 | <0.01 | <0.01 | <0.01 | <0.01 | 0.03 | 0.04 | <0.01 |
| Oxazolidinone/Phenicol (1) | 0.03 | 0.02 | 0.1 | <0.01 | - | <0.01 | <0.01 | <0.01 | - | 0.09 | <0.01 |
| Missing (26) | 0.31 | 6.65 | 0.17 | 1.13 | 19.4 | 1.34 | 10.77 | <0.01 | 0.9 | 2.83 | 0.2 |
| Macrolide/Tetracycline (1) | <0.01 | <0.01 | <0.01 | <0.01 | - | <0.01 | <0.01 | <0.01 | - | <0.01 | <0.01 |
| Macrolide/Streptogramin (4) | 1.69 | 1.1 | 0.12 | 0.39 | 1.01 | 1.18 | 1.43 | <0.01 | 1.83 | 0.48 | 0.11 |
| Macrolide (24) | 1.86 | 3.04 | 0.36 | 16.71 | 5.78 | 1.9 | 2.53 | <0.01 | 4.22 | 6.83 | 0.25 |
| Lincosamide/Streptogramin (2) | 0.12 | 0.28 | 0.02 | <0.01 | <0.01 | 0.01 | 0.38 | <0.01 | 0.06 | <0.01 | <0.01 |
| Lincosamide/Pleuromutilin/Streptogramin (7) | 0.48 | 0.27 | 0.09 | 0.02 | 0.18 | 0.02 | 0.34 | <0.01 | <0.01 | 0.21 | 0.05 |
| Lincosamide/Oxazolidinone/Phenicol/Pleuromutilin/Streptogramin (3) | 0.41 | 0.49 | 1.36 | 0.5 | 0.01 | <0.01 | 0.45 | <0.01 | 1.64 | 1.25 | 0.04 |
| Lincosamide/Macrolide/Streptogramin/Tetracycline (1) | <0.01 | <0.01 | <0.01 | <0.01 | <0.01 | <0.01 | <0.01 | <0.01 | 0.07 | <0.01 | <0.01 |
| Lincosamide/Macrolide/Streptogramin (44) | 6.12 | 12.33 | 14.51 | 2.26 | 1.35 | 0.32 | 17.71 | 0.03 | 8.19 | 9.52 | 5.95 |
| Lincosamide/Macrolide (1) | <0.01 | <0.01 | <0.01 | <0.01 | - | <0.01 | <0.01 | <0.01 | - | - | - |
| Lincosamide (9) | 1.88 | 3.33 | 3.42 | 8.29 | 29 | 0.36 | 2.02 | <0.01 | 4.01 | 11.61 | 0.71 |
| Glycopeptide (27) | <0.01 | 0.29 | 0.01 | <0.01 | - | 0.01 | 0.58 | <0.01 | 0.72 | 0.02 | 0.01 |
| Fosfomycin (28) | 4.36 | 1.07 | 0.2 | <0.01 | 0.01 | 1.29 | 1.49 | <0.01 | 0.22 | 0.03 | 0.19 |
| Folate_pathway_antagonist (37) | 2.92 | 3.2 | 2.19 | 0.41 | 0.97 | 5.45 | 3.13 | 0.36 | 4.33 | 1.03 | 0.82 |
| Beta_lactam (216) | 41.3 | 14.32 | 0.64 | 0.57 | 0.7 | 7.28 | 8.34 | 1.84 | 2.14 | 2.89 | 1.69 |
| Aminoglycoside/Fluoroquinolone/Quinolone (1) | <0.01 | 0.19 | 0.09 | - | <0.01 | 0.46 | 0.15 | <0.01 | 0.01 | <0.01 | <0.01 |
| Aminoglycoside/Fluoroquinolone/Macrolide/Phenicol/Rifampicin/Tetracycline (1) | 0.63 | 1.08 | 0.12 | 0.07 | 0.52 | 0.34 | 1.08 | 0.2 | 1.2 | 0.17 | 0.02 |
| Aminoglycoside (144) | 17.64 | 9.12 | 11.51 | 5.24 | 8.42 | 16.7 | 8.51 | 0.08 | 10.12 | 14.08 | 1.58 |

Source

Relative abundance: 75, 50, 25

**FIG 1** Relative abundance of read fragments aligned to each resistance class (row) for each sampling source (column). The relative abundance is the number of read fragments aligned to the group out of all fragments aligned to ARGs. For each resistance class, the parenthesis shows the number of ARGs belonging to that category.

There was a limited number of correlations for ARGs encoding resistance to fluoroquinolones, steroid antibiotics (fusidic acid), colistin, and rifampicin. On the other hand, beta-lactam, tetracycline, and aminoglycoside ARGs had many correlations with each other and with other classes (Fig. S4 and S6). Despite resistance to some antimicrobial classes being the most abundant, the ARGs did not always correlate to many others. For example, tetracycline ARGs were the most abundant in dog metagenomes, but no correlations were inferred for these ARGs. Similarly, in the human samples where aminoglycoside and beta-lactam ARGs were less abundant than tetracycline ARGs, aminoglycoside and beta-lactam resistance genes had a higher number of correlation coefficients were reported (Fig. 1; Fig. S4).

On the level of ARG abundance, we observed that just because an ARG was highly abundant in a sampling group, it did not automatically mean that it correlated to many other ARGs. The highly abundant *catA1* gene in marine (97.4% of all ARG reads), freshwater (55.1%), and soil samples (84.7%) (Fig. S2) did only correlate with one or two other genes in the water environmental networks and none in the soil network. On the other hand, *catA1* did seem to be correlated with 15 other genes in the pig correlation network despite not being highly abundant in that group of samples (Fig. S5a). *mef(A)_1* accounted for 15.9% of the reads aligned to ARGs in cow samples and 6.63% in pigs (Fig. S2) and was also strongly correlating with other genes, which mainly conferred resistance to aminoglycosides, (fluoro)quinolones, and tetracyclines (Fig. S5b). *tet(L)_4* only accounted for 4.01% of the read fragments aligned to ARGs in metagenomes of

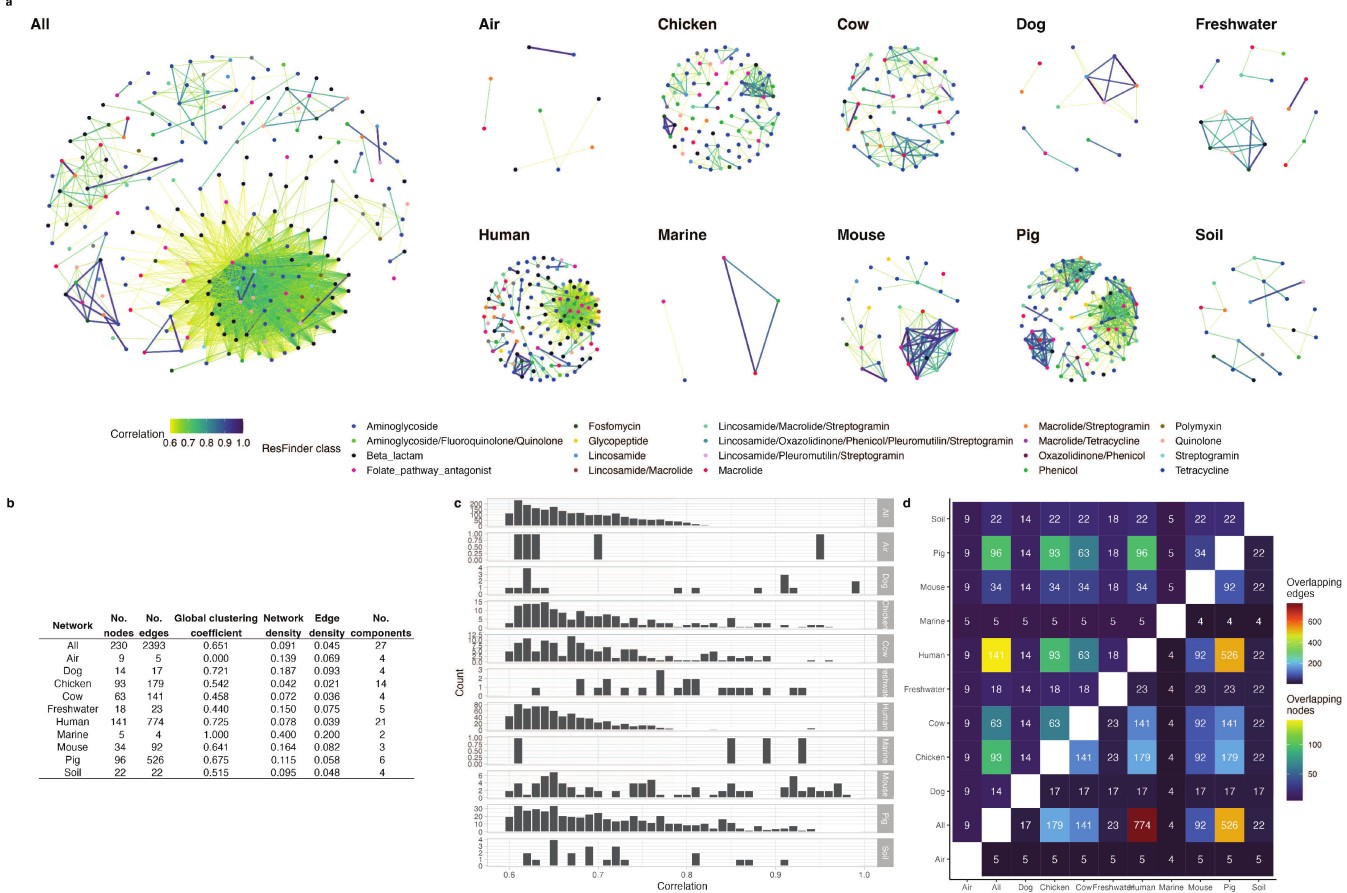

**FIG 2** The resistome networks consisting of correlations (edges) between pairs of ARGs (nodes). (a) Each correlation network is visualized, where each ARG node is colored by resistance class, and edges are colored by the correlation coefficient value. (b) Metrics of the correlation networks reveal how interconnected and complex the networks are. "No." stands for "number of". (c) Distribution of included correlation coefficients in each network. Figure S3 shows the distribution of all inferred correlations and their *P* values. (d) A heatmap showing how similar the content of the two networks is. In the upper half of the heatmap, the number of overlapping ARG nodes is shown. An overlap means that a specific ARG has a correlation in both networks, ignoring what it is co-abundant with. In the lower half of the heatmap, the number of overlapping correlation edges is shown. An overlapping edge is defined as whether the correlation coefficient is reported in both networks, regardless of the value of the coefficient.

chicken origins (Fig. S2) but was shown to correlate in its abundance with the abundance of 8 other ARGs, for example, with a correlation of 0.77 with *lnu(A)_1* (Fig. S5c).

## The hidden signals between ARG profile and the potential risk of co-selection in different environmental contexts

Using the correlations between ARGs in the various environments (Fig. 2), we calculated the average correlation between ARGs of different antimicrobial classes (Fig. S6). These average correlations can then serve as profiles to assess the risk of indirectly selecting ARGs that confer resistance to different ARGs through co- and cross-resistance. These risk profiles can then be used to judge the strength of interactions upon using one antimicrobial in each setting. Upon constructing these profiles, we observed that the strength and the number of correlations highly depend on the antimicrobial classes and the environmental context (Fig. S6). We hypothesize that if an important antimicrobial (IA) or highly important antimicrobial (HIA) is used, first, the resistance to the antimicrobial class will likely flourish, and, secondly, through co- and cross-resistance, so will ARGs conferring resistance to other classes, including those that are CIA.

Figure 3 shows two risk profiles for ARG correlations for glycopeptide and macrolide, two highly CIAs. Correlations between ARGs of glycopeptide resistance to other

resistance classes were much rarer than those connected with macrolide ARGs (Fig. S6), and those correlations that were observed were relatively low (correlation < 0.8; Fig. 3). Different vancomycin resistance cassettes were responsible in different environments, namely, *VanHAX*, *VanC2*, and *VanX_bc* in human samples and *VanHDX* and *VanC1XY* in pig samples (Fig. S7a). *VanHAX* only has one correlation, whereas the remaining five correlate with many different ARGs. On the contrary, ARGs conferring resistance to macrolide were much more interactive with other classes of resistance genes in all networks and had strong correlations with specific classes (correlation > 0.9; Fig. 3; Fig. S6). The macrolide ARGs co-abundant with other ARGs were many but separated into distinct network clusters (Fig. S7b). For example, *mef(A)* and *msr(D)* were usually found together in different environments, both in small and large clusters.

While Fig. 3 highlights how CIA ARGs interact, it is just as important to investigate what ARGs of less CIAs correlate with. As seen in Fig. 4, ARGs for pleuromutilin resistance (IA) and for tetracycline resistance (HIA) were found to interact with many other classes, including those that are critically important. For example, pleuromutilin ARGs are few but well connected (Fig. 4; Fig. S6), as seen with the connections with *lsa(E)* and *cfr(C)* (Fig. S8a). Tetracycline ARGs correlated with the abundance of multiple ARGs, such as those conferring resistance to lincosamides, macrolides, and phenicols (Fig. 4). While there were many ARGs for tetracycline resistance, they often correlated to the same ARGs (Fig. S8b).

## DISCUSSION

Considering the complexity of microbiomes, studying how microbial composition shapes the distribution of ARGs is a challenging task but one that could shed light on how ARGs indirectly select one another. However, with the high-throughput sequencing technologies and many metagenomic data sets available in public repositories, it is now much more feasible to extract the patterns of how ARGs co-occur without knowing their microbial origin. Using our recently published collection of 214K metagenomic datasets (27), we have inferred correlations of pairwise ARG abundances to profile which types of

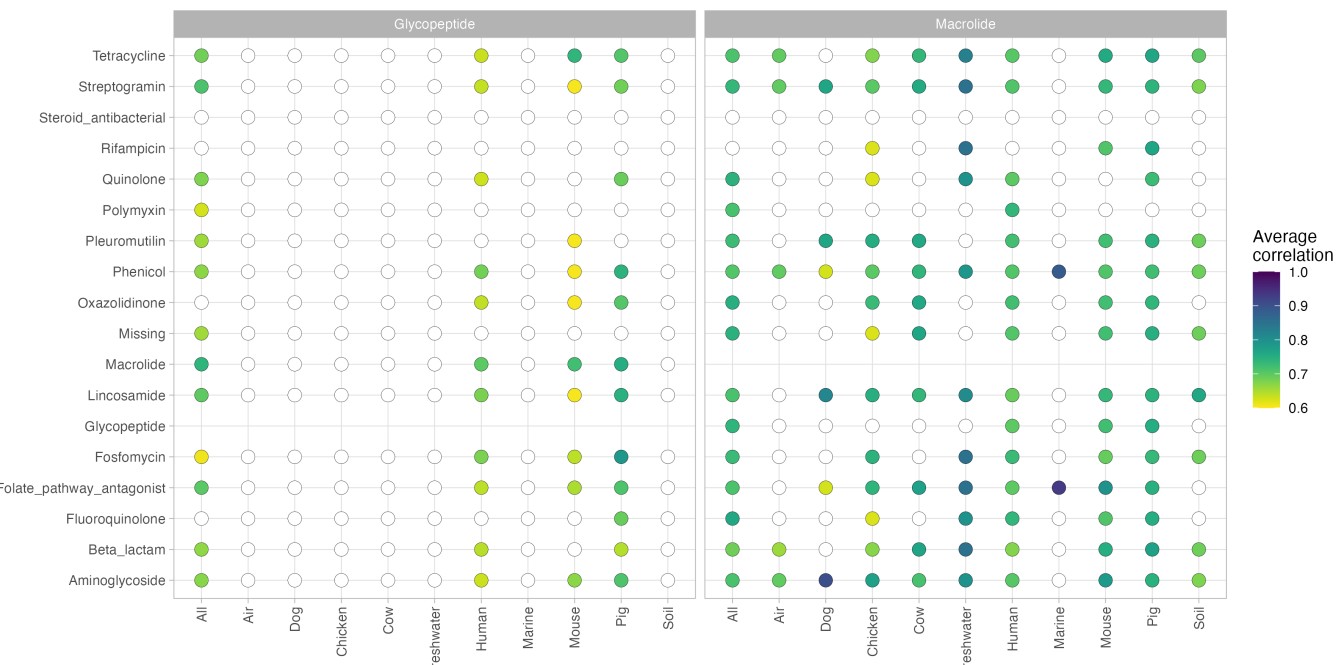

**FIG 3** Correlation profiles for ARGs conferring resistance to the CIA classes glycopeptides (left) and macrolides (right). Each column shows the average correlation from, for example, macrolide ARGs to ARGs for other antimicrobial classes. The circle is colored by the average correlation, where a white circle indicates no statistically significant correlations of ARGs observed between the two antimicrobial classes.

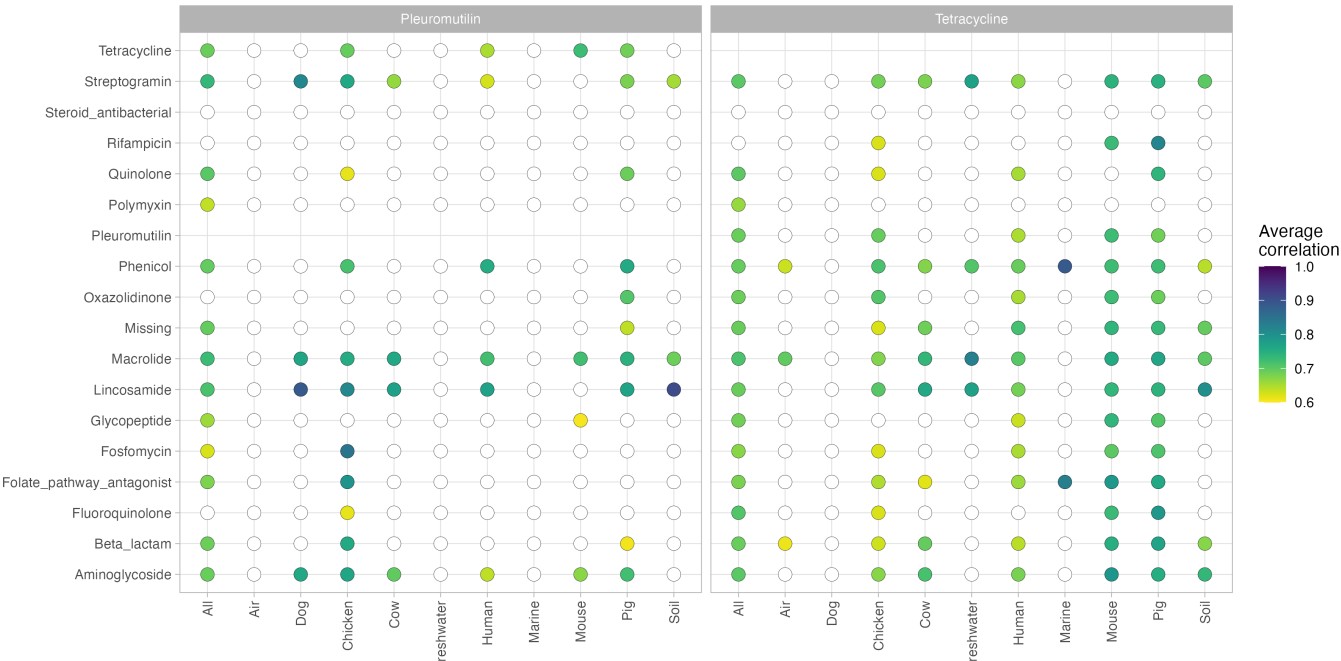

**FIG 4** Correlation profiles for ARGs conferring resistance to important pleuromutilins (left) and highly important tetracyclines (right). Each column shows the average correlation from, for example, tetracycline ARGs to ARGs for other antimicrobial classes. The circle is colored by the average correlation, where a white circle indicates no statistically significant correlations of ARGs observed between the two antimicrobial classes.

resistance influence the shape of resistome and which genes are the key players. To the best of our knowledge, this is the first study to relate ARG abundances on such a large and broad scale.

Our correlation networks revealed that not all ARGs are connected, as the pairwise ARG interactions were largely shaped by the composition of the environmental resistome. We observed cases where a highly abundant ARG had no correlations and the opposite where a sparse ARG had many correlations, highlighted by our effort to apply appropriate filters to ensure robust correlations (Table S1). However, by filtering on the individual abundance of ARGs in each resistome, there is a risk of removing important links between ARGs, especially in environments with very low abundant ARGs. We continued with settings that ensured that sparse ARGs in complex environments were removed but that ARGs in overall sparse environments could still be included, such as for the air resistome. More sensitive filtering could be applied to further improve the robustness of the inferred ARGs for each resistome.

The network showing correlations for all metagenomic samples was the most complex, which we speculate is due to both the wide variety of sampling sources and the overrepresentation of human metagenomes (Table 1). We found the differences in the animal and environmental networks much more interesting, as they reflect the dynamics of ARG abundances in more localized settings (Fig. 2). On the level of which antimicrobial class an ARG confers resistance to, we could also observe that some resistance classes had more and stronger correlations than others (Fig. S6). We found several cases of strongly correlated ARG pairs that have also been reported in other studies. These cases were often due to the ARGs being present in the same microbial genome. For example, the gene cassette *vanHAX* has been found together with *msr(C)* and *aac(6′)* in genomes of human isolates (48) and *mef(A)* linked with *tet(O)* (49) and *mdf(A)* with *blaTEM*, *aph(6)*, *sul2*, and *tet(A)* (50) (Fig. S7).

As highlighted in Fig. 3 and 4, we argue that the correlations can serve as a way to profile the collateral effects occurring in a resistome when the abundance of different ARG mechanisms changes. Antimicrobials have been classified differently to reflect their

importance to human medicine, of which glycopeptides and macrolides are CIA with the highest priorities. Glycopeptide and macrolide resistance have previously been linked genetically (7, 9) in pigs, where we also can report the presence of correlations between ARGs of glycopeptide and macrolide resistances not only in pigs but also in human and mouse environments (Fig. 3). Pleuromutilin and tetracycline antimicrobials are classified as being less critically important (51), but SparCC reported many correlations of ARGs for these two classes in various networks (Fig. 4), which suggests that there are still risks associated with the use of these two. Tetracycline resistance has been found to occur together with resistance to macrolides (52–54), aminoglycosides (52, 55), folate pathway antagonists (56, 57), lincosamide (11), and beta-lactams (55), to name a few studies. This high connectivity of tetracycline ARGs seems in line with our results, as this specific group of ARGs was connected to almost all classes of antimicrobials in most of our networks (Fig. 4; Fig. S8b). The variety of connections to antimicrobials of less importance to human health should be more in focus, as our results show that there are risks of critical AMRs emerging from the enrichment of less essential resistance genes.

Following this line of thought of focusing on ARGs that give resistance to less CIAs, a recent study by Tarek and Garner (2022) (58) proposed to create a monitoring framework based on isolated components, or clusters, in correlation networks. They constructed a correlation network encompassing ARG abundances in samples from wastewater treatment plants and argued that representative ARG members from each cluster in their network should be monitoring targets. Our networks suggest that limiting to only a few ARGs would fail to capture the complete picture in some environments, such as human microbiomes (Fig. 2). Instead, we propose to limit the set of monitoring targets to only what happens to ARG abundances during exposure to one type of antimicrobials (Fig. 3 and 4; Fig. S7 and S8). If a monitoring system is implemented, it would need to be updated regularly to show the changes in AMU and ARG co-occurrences since the correlations we have inferred in this study only reflect the current and past abundances.

In order to use correlation networks for surveillance of AMR, more work is needed to confirm that the observed interactions do indeed exist in nature (59). We have defined interactions as being indirect since SparCC does not indicate which way the interaction occurred. One way to investigate this could be by investigating physical linkages or using other analytical methods. For example, to determine whether *vanHAX* influences *msr(C)* or the other way around (Fig. S7a), the SPIEC-EASI (60) method could be used to infer such directional dependences. A directional correlation could be included in a risk profile.

Our choice of only studying the co-abundances of ARGs does not capture the causality and directionality of the interactions. We have not disentangled whether the interactions are due to co-selection or cross-selection and neither investigating genetic linkages, co-exposure, and the presence of other genetic elements such as mobile genetic elements. Understanding some of these factors could be achieved by expanding our analysis with bacterial read counts and counts of mobile genetic elements. We have speculated that the resulting correlations could reflect the community's response to external stressors, such as different usages of antimicrobials. While there has been work on linking the ARG prevalence with AMUs in various settings (7, 9, 11), incorporating AMU values, other non-antimicrobial stressors (61), and microbial signatures (12) in our analysis would likely help us better understand the microbiomes' response to selective pressures.

By utilizing the wealth of information on ARG abundances available in a collection of 214K metagenomic data sets, we have studied the co-abundance of ARGs to discover how these interactions shape the prevalence of resistances in different environments. The inferred correlation networks provide insights into how two resistance types indirectly and species independently select for each other in different habitats. Our results further highlight that there are instances of genes of one type of resistance often co-occurring with many other types of resistance and that the environmental context

plays an important role, revealing them as important targets in surveillance programs to limit their impact on global health.

## AUTHOR AFFILIATIONS

[1]Research Group for Genomic Epidemiology, Technical University of Denmark, Kongens Lyngby, Denmark
[2]Biosciences Department, Faculty of Sciences and Technology, University of Vic - Central University of Catalonia, Vic, Spain

## AUTHOR ORCIDs

Hannah-Marie Martiny http://orcid.org/0000-0001-6733-7888
Patrick Munk http://orcid.org/0000-0001-8813-4019
Christian Brinch http://orcid.org/0000-0002-5074-7183
Frank M. Aarestrup http://orcid.org/0000-0002-7116-2723
M. Luz Calle http://orcid.org/0000-0001-9334-415X

## FUNDING

| Funder | Grant(s) | Author(s) |
| --- | --- | --- |
| Novo Nordisk Fonden (NNF) | NNF16OC0021856 | Hannah-Marie Martiny |
| | | Patrick Munk |
| | | Christian Brinch |
| | | Frank M. Aarestrup |
| | | Thomas N. Petersen |
| EC \| H2020 \| H2020 Societal Challenges (SC) | 874735 | Hannah-Marie Martiny |
| | | Patrick Munk |
| | | Christian Brinch |
| | | Frank M. Aarestrup |
| | | Thomas N. Petersen |

## DATA AVAILABILITY

The matrix of raw ARG read counts is available at https://doi.org/10.5281/zenodo.6519843 . The code used to run the analysis and create figures and the SparCC output files are available at https://github.com/hmmartiny/global_resistome_correlations. Classifications of antimicrobial importance were retrieved from the 6th revision of critically important antimicrobials (CIAs) for human medicine from https://www.who.int/publications/i/item/9789241515528, accessed 10 October 2022.

## ADDITIONAL FILES

The following material is available online.

### Supplemental Material

**Supplementary material (Spectrum04108-23-s0001.pdf).** Table S1; Fig. S1 to S8.

### Open Peer Review

**PEER REVIEW HISTORY (review-history.pdf).** An accounting of the reviewer comments and feedback.

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
