## [Reviewer comments · Microbiology Spectrum]

Microbiology Spectrum

Utilizing co-abundances of antimicrobial resistance genes to identify potential co-selection in the resistome

Hannah-Marie Martiny, Patrick Munk, Christian Brinch, Frank Aarestrup, M.Luz Calle, and Thomas Petersen

Corresponding Author(s): Hannah-Marie Martiny, Danmarks Tekniske Universitet

Review Timeline:

Submission Date:	December 11, 2023
Editorial Decision:	March 8, 2024
Revision Received:	April 5, 2024
Accepted:	April 19, 2024

Editor: Monica Garcia-Solache

Reviewer(s): The reviewers have opted to remain anonymous.

Transaction Report:

DOI: <https://doi.org/10.1128/spectrum.04108-23>

Re: Spectrum04108-23 (Utilizing co-abundances of antimicrobial resistance genes to identify potential co-selection in the resistome)

Dear Dr. Hannah-Marie Martiny:

Thank you for the privilege of reviewing your work. Below you will find my comments, instructions from the Spectrum editorial office, and the reviewer comments.

Revision Guidelines

Sincerely,
Monica Garcia-Solache
Editor
Microbiology Spectrum

Reviewer #1 (Comments for the Author):

Dear Author

Thank you for your manuscript submission. The manuscript is well-designed and well-presented; however please do revise as bellow:

1. Please do revise some typos and grammatical errors.
2. Please do add some references regarding the used protocols

We appreciate the time the reviewer took to comment our manuscript and have incorporated their suggestions.

According to the first request, we have revised typos and grammatical errors in the following lines, which have been highlighted in red in the main manuscript lines: 65-66, 116, 146, 165, 332-333, 343, 347, 472-473.

For the second request, we have clarified our protocols with additional references in the following lines, which are highlighted with blue in the manuscript: 97-103 and 110.

Re: Spectrum04108-23R1 (Utilizing co-abundances of antimicrobial resistance genes to identify potential co-selection in the resistome)

Dear Dr. Hannah-Marie Martiny:

Your manuscript has been accepted, and I am forwarding it to the ASM production staff for publication. Your paper will first be checked to make sure all elements meet the technical requirements. ASM staff will contact you if anything needs to be revised before copyediting and production can begin. Otherwise, you will be notified when your proofs are ready to be viewed.

Sincerely,
Monica Garcia-Solache
Editor
Microbiology Spectrum